# A Psychoanalytic Approach to Internet Gaming Disorder

**DOI:** 10.3390/ijerph20156542

**Published:** 2023-08-07

**Authors:** Georgios Floros, Ioanna Mylona

**Affiliations:** 12nd Department of Psychiatry, School of Medicine, Aristotle University of Thessaloniki, 56430 Thessaloniki, Greece; 2Department of Ophthalmology, General Hospital of Serres, 62100 Serres, Greece; milona_ioanna@windowslive.com

**Keywords:** internet gaming disorder, self psychology, psychoanalysis, guided imagery, bipolar disorder

## Abstract

Background: Internet Gaming Disorder (IGD) is now an official diagnosis and significant public health challenges have been already identified regarding the provision of appropriate care to patients of all ages and the preparedness of mental health professionals to manage the disorder. Despite the existence of psychotherapeutic treatment modalities available for some time now, there is a paucity of any psychoanalytically driven treatments and the disorder is widely regarded and classified as being ‘behavioral’. This has profound implications for patients with long-standing character pathology and psychiatric comorbidities, who are underserved by the provision of health services that could efficiently address their issues. Methods: This study presents a psychoanalytic perspective on IGD, based on Kohut’s Self Psychology as applied in the treatment of other addictions. An outline of the theory, assessment and treatment modalities is presented with two case reports that illustrate its application. Results: The presentation outlines the challenges in treating IGD, expanding on the concept of guided imagery, resistance to treatment, selfobject transference and comorbidity with marijuana use and bipolar disorder. Conclusions: A psychoanalytically driven protocol can be effective in treating IGD, especially in cases with marked character pathology and low motive to engage in other treatment modalities.

## 1. Introduction

Internet addiction (IAD) as a term entered the scientific discourse more than twenty years ago [1], with addiction to online digital gaming (Internet Gaming Disorder—IGD) rapidly becoming the prime example of this disorder, leading to its preliminary inclusion in the DSM-5 [2]. IAD has been difficult to define due to the all-encompassing term ‘Internet’ that includes all possible online activities; a simple working definition that was proposed [3] is that IAD should be diagnosed in those Internet users whose use has become problematic or pathologic, specifically because of their inability to control it despite adverse life consequences. The fact that the vast majority of IAD cases revolved around online video gaming led to the more limited entity of IGD becoming the focus of recent research. IGD was defined in DSM-5 in accordance to established criteria for pathological gambling and this definition largely persisted due to its simplicity, with five out of nine possible overt signs and symptoms denoting pathology. The list of symptoms included: preoccupation with gaming, withdrawal symptoms, increased tolerance as evidenced by more time spent gaming, unsuccessful attempts to control gaming, loss of interest in previously valuable offline activities, continued use despite knowledge of the issue, deception of family members, therapists, or others regarding the amount of gaming, use of games to escape or relieve negative mood, and jeopardizing or losing a significant relationship, job, or education or career opportunity because of gaming. The consistent rise in reported cases led to ‘Gaming Disorder’, including both online and offline gaming, recently included as a distinct entity in the 11th Revision of the International Classification of Diseases (ICD-11) [4] under the umbrella term of a ‘behavioral addiction’. This classification is shared along with pathological gambling, long considered a prototypical behavioral addiction. The ICD-11 criteria are more open-ended and focus on three essential features: impaired control over gaming, gaming taking precedent over other activities and its continuation or escalation despite negative consequences. A recent comparison between the proposed criteria in the DSM-5 and the ICD-11 concluded that the ICD-11 diagnosis emphasizes serious symptoms, such as functional impairment caused by excessive gaming over a long time, and it supported the validity of ICD-11 diagnosis for future use [5]. A recent study exploring what are the attributes of online preoccupation which may lead to addictive use of the medium [6] concluded that there is a number of key factors associated with IAD and IGD: disinhibition, ease of access, content stimulation, synergistic amplification of the effective internet delivery mechanism with the content, boredom intolerance versus the always on nature of the Internet, dissociation and loss of time perception, perceived anonymity, and the variable activation of neurobiological reward pathways shared by other addictions to psychoactive substances and behaviors.

Online gaming per se has seen a meteoric rise throughout the last decade. Comparative data have shown a doubling of the number of online gamers worldwide during 2017–2023 to 1.07 billion gamers, with a revenue of $26.14 billion [7]. The most lucrative game genre by far is that of the MMORPG (massively multiplayer online role-playing game), accounting for $23.35 billion in revenue. This genre includes story-driven online games, in which a player, taking on the persona of a character in a virtual or fantasy world, interacts with a large number of other players, creating short-lived or longer-standing alliances. Revenue for the game companies can be either directly driven from selling the game, selling a subscription to the game or, in the so-called ‘free-to-play’ games, making the game available for free but charging the players for any extra bonuses in gameplay via ‘micro-transactions’. A ‘micro-transaction’ is an in-game purchase of relatively small monetary value that seemingly offers disproportionately larger value within the game itself [8]. A related notion is that of the ‘loot box’, a system of random gains within the game which can be unlocked either by hours of tedious gameplay or by a micro-transaction, thus effectively mimicking the mechanics of online gambling [8].

Prevalence of IGD has been difficult to establish during the past decade due to the lack of a widely accepted definition and appropriate methodology. The most recent meta-analysis by Kim et al. [9] estimated a prevalence for IGD of 2.4% (95% CI 1.7–3.2) when examining only those 28 studies that had a representative sample of the examined population. A review of longitudinal studies found a wide variance in the temporal stability of the diagnosis, ranging from 20 to 84% depending on age and study duration, with most studies limited to a single year and relating to minors [10].

The implications of this new phenomenon for public health were quickly pointed out by experts in the field of both Public Health and Addiction Psychiatry; Professor Dimitri Christakis, after describing a case of a 28-year-old Korean who suffered a fatal cardiac arrest following a 50-h-internet gaming session, highlighted a general complacency over the issue and claimed that it could evolve into a 21st century epidemic [11]. Current (2023–24) president of the American Psychiatric Association, Professor Petros Levounis has issued what is being described as a ‘a wakeup call alerting the medical community—and society at large—to the addictive potential of technology’ in his recent work, ‘Technological Addictions’ [12], co-authored with James Sherer. Both authors stated in a recent related review of published material that society’s dependence on addictive technologies, including but not limited to IGD, will only increase [13]. They stress the case that technological addictions should be viewed as legitimate psychiatric conditions worthy of medical assessment, diagnosis, and treatment. With online gaming a favorite pastime of millions, the existence of even a small percentage of affected players immediately translates to a major public health issue.

The acceptance of IGD as a valid diagnosis, according to Long et al. [14], highlights a number of global healthcare challenges, including increased care needs among affected people and their relatives and health professionals’ preparedness to accurately identify and manage gaming disorder. The authors suggest that appropriate clinical management and treatment should be further developed and delivered with accumulating evidence from clinical practice [14]. Ideally, treatment options should be tailored to the individual, since IGD has a broad range of affected individuals with regards to gender, age, and pre-existing psychiatric needs or co-morbid psychiatric disorders. However, although treatment options are available for some time now [15,16], there has been a dearth of psychoanalytic treatment modalities available, with most treatment options following the cognitive-behavioral paradigm and the remaining options being either family-systemic or even pure behavioral group programs [17,18]. Although cognitive-behavioral and family therapy treatments may be more relevant in younger age, when character defenses and pathology are still fluid, a gap is evident in clinical practice for patients who have been engaged in addictive gaming for the better part of their lives. While online gaming initially was primarily a favorite pastime activity of young adolescents, its persistence in time led to a gradual expansion of the average age of gaming up to 35 years of age in the US and elsewhere. During a 2022 survey, 36% of video game players still come from the 18 to 34 age demographic, and 6% are 65 years and older [19]. A meta-analysis of affected game players found prevalence rates for adults that were comparable to those of children and adolescents [20]. Hence, a need for treatment modalities relevant to adults with character pathology and other psychiatric comorbidity has become more acute as time passes by and is largely unheeded by programs tailored for younger ages.

A significant obstacle in the provision of psychoanalytically guided treatment for IGD is the fact that there is very little in the way of psychoanalytic theory for online gaming disorder despite some early pioneering work on interaction with computers as early as 1984, with Turkle noting from a Lacanian viewpoint the existence of computer users who saw interaction with the machine as a goal in itself rather than a means to a goal [21]. A recent in-depth review of theory and treatment of IGD [22] completely overlooks any analytic contribution in the study of addiction in general, referencing only a single article by Taipale [23], yet failing to expand on its content. This is unfortunate since Taipale aptly highlights a core issue with all ‘behavioral’ addictions: the medium of addiction by itself cannot be considered as ‘addictive’ since only a minority of users become addicted, then it is ‘..more reasonable to search for the origin and cause of addiction in the structures of the experiencing subject’ ([23] p. 30). Interestingly, while various theories of personality have been researched over the years with regards to IGD and IAD in general [24], their impact on case formulation and treatment has been negligible. There is very little basic research in this respect from an analytic perspective. Floros et al. [25] have examined whether character defenses may assist in differentiating internet-addicted college students from controls and found that maladaptive defenses were more common with addicts; specifically Internet addicts scored higher on denial, fantasy, help-rejecting complaining, isolation, and lower on sublimation, compared to controls. Furthermore, the comorbidity of IAD with personality disorders reached up to 38% of all cases among adult college students who underwent treatment for IAD, with the majority (11 out of 19 cases diagnosed with a personality disorder) being that of narcissistic disorder [26].

Various treatment services have been setup for IGD, long before its official recognition as a psychiatric disorder. This study will outline the practice in one of those centers, the ‘student outpatient Service for the treatment of problematic use of personal computers and the internet’ that was founded in 2011 [15]. Patients without significant character pathology or with high motive for change typically follow a brief, cognitive-behavioral treatment plan. However, it was soon evident that up to 40% of all patients have significant character pathology [26] and a more in-depth approach was deemed necessary in the majority of those cases. This manuscript will illustrate the approach that was employed, aiming to provide data from clinical practice that will be relevant in setting up comprehensive treatment services in order to address the needs of an over-looked patient population, that of the adult patient with long-standing addictive gaming and character pathology, with or without psychiatric co-morbidity.

## 2. Materials and Methods

### 2.1. Aim of This Study

This paper will attempt to expand on the psychoanalytic perspective of addictive online gaming with a parallel presentation of clinical cases. The foundation of this perspective lies in the depth psychology of the Self, as formulated by Heinz Kohut [27] during the late 1970s, and expanded to a psychoanalytic study of addiction by Ulman and Paul [28]. However, the treatment plan was greatly assisted by the adoption of the therapeutic modality of guided affective imagery (GAI) [29]. GAI is the verbal evocation of waking dreams, where the patient is instructed to mentally explore a particular fantasy setting, in which he/she effectively explores aspects of the unconscious and interacts with meaning-laden symbolic objects. This process was well-received in most instances by patients and helped immensely with the establishment of a therapeutic alliance. The average gaming-addicted patient approaches therapy with ambivalence, denial and essentially rejecting outside help, as confirmed by earlier research [25]; the GAI setting offered a treatment pathway where important details of the inner life are brought fore that would otherwise require a very lengthy consultation period for the patient to offer spontaneously, if at all. Patients who were up to that point unwilling to verbalize their inner conflicts found a gateway that appeared safe and largely eluded their unconscious defenses, much in parallel to working with dreams. Moreover, gaming-addicted patients have a penchant for daydreaming and especially escapist fantasy life [30], as directly manifested by their predilection for fantasy-driven role-playing online games, a fact that bodes well with a number of specific GAI scenarios.

### 2.2. Theoretical Background

The analytic framework for this investigation lies in the work of Kohut on the narcissistic disorders of the Self. Kohut discarded the Cartesian dualism in Freud’s original theoretical framework, where there are clear boundaries between the Ego and the external objects, for a new scheme where the complete Self is derived and fueled by a constant internalization of the experiences of the subject-in-relation to the object and the object-in-relation to the subject. The selfobject was introduced as a construct that includes the dimensions of our experience of another person that relates to this person’s functions in establishing our sense of Self; the Self is constructed in childhood through confirmation from the primary caregiver, as it vacillates between grandiosity and the gravitas of an idealized parental image [31]. It is important to note that a selfobject may not necessarily be human in its origin [32]. A nonhuman selfobject may be invested with libidinal energy as well, offering an additional dimension of being under the total control of the developing subject, thus boosting the grandiose Self. This nonhuman selfobject (a physical object or an activity) in normal development, where the selfobject milieu of the caregivers is appropriate, is a transitional point of cathexis for libidinal energy which will be reinvested in other selfobjects, human or non-human, in normal play and social interaction. However, in cases of an empathic failure, or even downright antipathy from the caregiver, the nonhuman selfobjects are overinvested with libidinal energy and the child remains arrested developmentally in this phase of development. Here, the subject is arrested in a psychic status of unchecked grandiosity where control over the selfobject is valued above else and is in turn reaffirming of the subject’s grandiose self since it sorely lacks the inevitable optimal failures of the human caregiver selfobject that are structure-building [32].

Ulman’s and Paul’s work with addiction in a Kohutian framework centers around the existence of an addiction-prone personality, which has in the past faced the failure of the primary caregivers to form a structure-building selfobject in the psychic apparatus of the child [28]. This led to a developmental arrest, due to the compensation for this lack with the ‘over-investment of narcissistic capital in nonhuman, or at best semi-human, things and activities. Gradually this solidifies in a specific type of archaic narcissistic megalomaniacal fantasy of possessing magically endowed things and activities, what the authors labeled as ‘addictive trigger mechanisms‘, or ‘ATMs’. The ATMs are the psychoactive agents or activities (with the authors purposefully not making a distinction between the two categories) which the subject feels that can yield in order to control both the reality of the sense of one’s self and one’s personal world. Interestingly the authors explicitly mention that ‘an addict imagines being like a sorcerer or wizard who controls a magic wand capable of manipulating the forces of nature, and particularly those forces of human nature’ ([28] p. 6). While this megalomaniacal fantasy is a phase-appropriate aspect of narcissism in its normal development from archaic to mature forms, as the child seeks ‘control over a narcissistically experienced world’ ([33] p. 393) which relieves stress through narcissistic bliss, the adult with an arrested development in this transition will actively seek to re-experience these early moods (affect and sensation) of narcissistic bliss through the use of ATMs, trapped into an addictive, megalomaniacal narcissistic fantasy. Chasing this fantasy leads the addict in a downward spiral where the decline experienced in the real world will necessitate an even deeper involvement with the fantasy world, regardless of the true personal cost, since there are no appropriate psychic structures in place to support this crumbling Self. The fantasies are part conscious and unconscious, in what Ulman and Paul termed a ‘hypnogogical state of consciousness’ ([28] p. 16); the authors suggested that through analysis the patient may be able to gain an understanding on the nature of these fantasies and work through them therapeutically, as they are revived in the form of a transitional selfobject transference with the therapist.

Kohut proposed in the treatment of narcissistic disorders that the therapist should empathically experience the world from the patient’s point of view, so that the patient feels understood [27]. The inevitable empathic failures of the therapist, however small in objective terms, will be experienced deeply by the patient and should be interpreted to help the patient understand the need to restore closeness and solidarity with the therapist. Shedding light in the megalomaniacal fantasy that the patient harbors is thus as important as is difficult, since the patient only partially understands it, on a cognitive level, and may be reluctant to be forthcoming with it. This reluctance in the average patient would stem from a difficulty in establishing trust in relationships; in an addictive disorder it is further exacerbated by only partial willingness and low motive to enter therapy and a typically long history of contact with psychiatric services and accumulated negative selfobject experiences that this entails. The addict often seeks the company of other addicts, as they form an accepting selfobject milieu [28], in which they all share parallel trajectories in their chase of their particular fantasy.

### 2.3. Case Assessment

All cases were followed in the ‘Student counseling service for problematic use of the Internet and personal computers’ of the Aristotle University of Thessaloniki, Greece, operating since 2010 [15]. During the initial assessment, the patient provides a personal history while a battery of diagnostic tests is offered for completion. A full psychiatric assessment for comorbidities is essential in order to identify potential underlying disorders. The psychometric tests include the proposed DSM-V diagnostic criteria for IGD [2] and Bond’s Defense Style Questionnaire (DSQ-88) [34]. The latter is a self-report scale that assesses unconscious defenses, clustered in ‘defense styles’. ‘Defense styles’ are describing the subject’s ability to adapt in challenging situations maturely rather than revert to immature behavior. Four defense styles are assessed: the ‘maladaptive style’ which is characterized by the employment of immature defense mechanisms, the ‘image distorting’ style characterized by the presence of primitive defense mechanisms, the ‘self-sacrificing’ style characterized by more mature mechanisms focused on social acceptance and inclusion, while the ‘adaptive style’ is characterized by mature defense mechanisms that channel drives to age-appropriate interests and goals.

### 2.4. Treatment Plan

The following notes will refer to conclusions drawn during the treatment of individuals with IGD, unless specifically referenced to another source. Typically, the patient will enter therapy driven by necessity, either having realized that his actions (or inaction) is self-defeating, or being coerced by significant others; in either case the patient explicitly seeks a fast improvement of his predicament and a better understanding of his otherwise inexplicable behavior. In a more traditional setting, the principal aim would be to establish the nature of the transference as it will unfold [27]. In this case there is a very fast progression; indeed, the longer the duration of the symptoms, the faster the transference will appear and with higher intensity. Unfortunately, the patient will show little motive for analytical work with the therapist on the nature of this transference, since any attempt to work it through will be met with hostility and denial, as is often the case when transference is established early on the therapeutic setting. The mere process of interpretation will be perceived as an aggressive gesture aiming towards the patient. Hence, the preferred course of action is to set in motion the guided imagery work as soon as possible once basic trust and rapport has been established and let the transference unfold in a more paced timeframe, observe it, yet refrain from commenting on it. As suggested by Leuner [35], transference does not play a pivotal role in GAI and that attribute is very useful in this setting. The first guided imagery scenario is a diagnostic exploration of the patient’s inner self via either the drawing of an imaginary tree or the exploration of a house, or both. These images are representative of the patient’s perceived sense of self to a different extent, with the drawn tree denoting the current emotional state and sense of relating to family and others, while the house imagery offers a view of the inner self, compartmentalized in the respective rooms as pioneered by Freud [36]. With regards to the draw-a-tree exercise, there are several metaphors that must be explored regardless of whether they are actually drawn or whether the patient is instructed to ponder their imaginary presence. The position of the tree on the ground relates to the current sense of foundation for the Self, the roots to the relationships with family and origins in general, the trunk to the actual Self, branches, leaves and fruit respectively to goals, activities and achievements, the sky to the current emotional state and the surrounding environment to the feeling of belonging. The patient can be encouraged to imagine an alternate setting in which the tree would be optimally placed. In the house GAI exercise, the external image and surroundings along with a detailed inner exploration are interpreted symbolically. Special attention is paid to the attic, representing childhood memories, and to the basement, representing the unconscious, with the patient instructed to explore and examine any objects of interest in these rooms.

In both instances these two exercises offer a fast pathway into the current emotional issues and perception of self, an important caveat since the patient will approach therapy reluctantly and with little interest for self-disclosure. Since the therapist demonstrates an active interest in understanding the inner world of the patient during the process, this also helps establish a selfobject transference early on, while maintaining analytical distance. The therapist should offer his/her impressions on the patients’ imagery, be open to their suggestions as to the personal meaning of the evoked symbols, validate their feelings related to the evoked images, and crucially, refrain from pushing interpretations aggressively. Further GAI themes for future sessions include exploring a river, signifying the life pathway, climbing a mountain that relates to the strive for achievement, and more advanced scenarios where the patient is instructed to specifically tackle challenges that relate to inner conflicts. In all instances, a debriefing session follows where the patient explores the meaning of each scene with the therapist who will offer a detailed discussion on the analytic meaning of each element in the scenario drawing on the rich analytic tradition already at hand. Special attention should be paid to how the patient perceives these elements and this perception should be validated against any symbolism. Expanding the field of associations from a scenario will provide useful material [35] and the patient will be more likely to acknowledge parallels with past and present motifs.

## 3. Results

Two cases will be briefly outlined to demonstrate the main points of treating patients with IGD under a psychoanalytically guided treatment protocol making use of guided imagery. These cases have been chosen to expand on challenging issues for the therapist that are often encountered in adult patients with IGD including treatment resistance and psychiatric comorbidities.

### 3.1. Case 1: George

George, a 25-year-old college student, came in for treatment following treatment of two years with supportive psychotherapy, mostly focused on handling his emotions and increasing motivation. Despite his treatment, George found himself unable to progress, having half-dropped out of his engineering studies, limiting himself to bouts of gaming, online scuffles on politics with strangers, binge-eating and smoking marijuana. George had very little motive for change and his treatment was initiated at the insistence of his parents, a divorced couple comprising a civil protection officer and an engineer. George lived with his mother, a workaholic and regular alcoholic, who secretly drank herself to sleep more often than not, and was prone to outbursts of helpless anger against her son; she could only calm herself with a considerable dose of alcohol. She would not, however, ask for help or openly admit her ‘secret’, which was known to her son and ex-husband. George’s father was a larger-than-life character, commended for his actions on multiple occasions. The couple divorced when George was eight and the father soon moved away, formed a new family, and fathered another son with whom George had a very caring relationship. George’s father took an interest on the well-being of his son, would regularly invite him at his new home, and tried to motivate him, with little to no effect.

George had an otherwise unremarkable childhood. He would study his school lessons only under pressure from his parents and with tutoring, but managed to pass his university examinations on sheer brainpower and little preparation. However, he soon became disillusioned with the way the curriculum was organized and was particularly irritated with what he perceived as unjust decisions and practices by his professors. He started skipping classes and fell back on his studies. He acknowledged that he was stalled and opted to pause and serve a mandatory tour-of-duty with the armed forces to help him change his pattern of inactivity. He completed a one-year tour without difficulties, the most notable change was him taking up marijuana use. Upon his return, he failed to make a comeback to his studies and also had a disappointing end to a fledgling romantic relationship. He appeared to sink into a mundane existence of online gaming, unhealthy eating, and low social functioning. After nearly three years of trying to get back on track, his mother contacted the outpatient service for some help regarding how she could best help him. Later, George was convinced to try out therapy, although his optimism was very low, following the end of his previous treatment which was concluded on his own initiative since he felt that he did not receive any meaningful help.

Although George spent a considerable amount of time online, by the time he had started therapy his ‘proper’ online gaming with MMORPGs and online shooters had waned down since most of his friends, who were also his teammates, were now employed and with little time to spare. He was more prone to spectate other players play and waste the rest of his time on online arguments on social media, with unknowns, getting involved in toxic online exchanges on issues that were morally important to him. He became a recluse, stopped all physical activity, used marijuana, and indulged in unhealthy eating.

George had average DSQ scores on the maladaptive, self-sacrifice and adaptive defenses, and higher than average on the image-distortion scale. He scored high on the consumption, denial, fantasy, help-rejecting complaining, omnipotence, splitting and undoing defenses. He also employed more mature defenses, such as humor, sublimation, task-orientation, pseudo-altruism, and anticipation.

What was more remarkable about George’s clinical course was the conscious resistance to the imagery sessions and lack of trust towards the therapist up till a considerable passage of time. George did not disclose his regular habit of smoking marijuana for more than fifteen months into therapy, a fact he had also successfully kept hidden from his mother, despite sharing an apartment with her. He would also actively sabotage a number of more involved imagery sessions by refraining from providing important details during the procedure or even lying initially about what he experienced, thus derailing the process. On an imagery session where the subject visits a swamp and essentially waits for a subconscious fear to emerge from within in order to face it, George initially offered a simple image of a frog, only to admit a few minutes later that in fact the figure that emerged from the depths was an elite military diver in full battle gear; the figure glanced at him with contempt, depicting an inner unforgiving and very critical aspect of his Self; George would always compare himself to, and inevitably fail to measure up to, an idealized internalized imago of complete success and dominance. His attempts at restarting his stalled life mirrored this, with George setting high goals, being very unappreciative of what he could achieve and procrastinating to avoid failure and the resulting sense of inner contempt for himself. George’s notable resistance during therapy was an early indicator of his narcissistic need that would manifest itself clearly during the selfobject transference between him and the therapist despite his extensive use of what Ulman and Paul noted as ‘ersatz selfobjects’ [28], namely gaming and marijuana use. As Kohut noted, the type of narcissistic transference points to the unmet selfobject need of the patient; in this case George clearly experienced grave concern and anguish over his mother’s state, as if still waiting on something he never received; gradually this unmet need manifested in a mirroring transference, expecting the therapist to confirm him. The dyadic relationship between George and his mother led to a vicious circle of him failing to progress, which seemingly exacerbated his mother’s drinking issue, which in turn seemed to make George even less likely to move forward. Both used ersatz selfobjects to essentially avoid working through their deeply rooted issues. George’s mother had herself a history of sexual abuse by her father, yet any attempt to motivate her into seeking treatment for her drinking issue was met with fierce resistance and anger.

A point of great importance, was the marked deterioration of his presentation when he had his one and only failed attempt at a romantic relationship five year before treatment. This in fact is a recurring incident of numerous patients with IGD and needs some elaboration since the classical approach would be to ascribe it to an Oedipal conflict. When placed in a guided imagery scenario of visiting a witch, evocative of aspects of the relationship to one’s mother, George found himself a guest of a benevolent witch, eager for a sexual encounter later that night; following that intercourse George found himself transported outside, staring at a starry sky, and feeling a majestic union with the entire universe. At that point he confided his inner fantasy of greatness equal in essence to a supreme being, capable of holding everyone’s lives at one’s hands. However, this was not the result of the sexual encounter but rather a pre-requisite; George needed to prove himself in the eyes of his mother in order to receive affection, yet her gaze was not following him during his early childhood but her own ersatz selfobject. George in this guided imagery scenario vividly demonstrated how his failure to progress past the Oedipal stage stemmed from his serious narcissistic pathology. When faced with his single episode of failure at his real-life romantic overtures, George regressed back to this stage of stunted development and rejected any notion of making another attempt, despite the fact that his failed attempt was essentially just a casual flirt. He remained fixated at that young woman and ‘followed’ her on social media ever since, reminding himself from time to time of his failure at what was clearly a self-punishing ordeal. The key for his recovery was the break-up of the unhealthy dyadic relationship, not only by confirming George, but also by motivating him to become financially independent and move out, following essentially the course of action that his father took when he realized that he could not help his ex-wife. This had the effect of depriving his mother from her single reason that she consciously attributed her drinking issue to; she eventually entered therapy on her own. George needed to work through his guilt of abandoning his mother, much as what he felt that his father did. The intense guilt of abandoning a caregiver, who appeared to need care more than she was able to provide, had shunted his own growth with a huge burden placed on his shoulders very early on his youth.

### 3.2. Case 2: Paul

Paul, a 23-year-old college student, came in for treatment at the insistence of his parents who sought out a specialist on IGD; Paul was under combined psychopharmaceutical and psychotherapeutic treatment for three and a half years with three separate psychiatrists, following two attempts to commit suicide while on a depressive episode. He was diagnosed with bipolar disorder and had received in the past various combinations of antipsychotic, antidepressant, and mood-stabilizing drugs with mixed results. He was currently on quetiapine XR, fluoxetine, and lamotrigine, and received regular consultations with a CBT therapist, yet his parents were worried that their son had lost every interest in life and only spent his time gaming, having failed his college exams repeatedly and refused to further follow any classes. Paul reluctantly agreed to meeting yet another therapist, making it clear that he had no intention of rejoining the ‘real life’ (RL). He saw his game playing as the only thing that made his life tolerable. Two years ago, Paul had a first manic episode followed by a depressive one, accepted treatment, then a few months later stopped his medication and went abroad for studies, believing that it was a single episode and not a full-blown disorder. A year later, during a second manic episode, Paul had a feat of rage and jealousy against his girlfriend that culminated with him physically assaulting her. He fell into a deep psychotic depressive episode with intense guilt, made an attempt on his life, and was hospitalized. This period of time was deeply traumatic for him; seeing it as proof that he was ‘defective’, would never be able to achieve his goals, and more importantly, feeling intense guilt, he declared himself unfit to live and presented his parents with his ultimatum: they would either allow him to game his life away or he would take it himself. He also smoked marijuana on a regular basis, as long as the meagre funds he received from his parents enabled him to purchase it.

Paul was the elder child of an unhappy marriage. His mother had a tempestuous character with emotional lability, outbursts of anger, and unpredictability. She had a tendency to force her current opinion onto others and would enforce corporal punishment for minor mishaps to her two children. The wedding ended in divorce and Paul’s mother came out as a lesbian, trapped in a life that did not express her. Her newly assumed sexual orientation did little to stabilize her emotional state and her erotic relationships were characterized by outbursts of emotion during regular breakups and rejoining’s. Paul’s father was a successful businessman who, following the divorce, left his two children with his ex-wife, moved away, and remarried. Paul lived with his mother, in a tense relationship, with frequent arguments.

Paul had low DSQ scores on both the maladaptive and adaptive defenses, and also on the self-sacrifice scale. Image-distortion scores, however, were considerably higher than the normative scores. He scored high on the denial, help-rejecting complaining, isolation, withdrawal and regression defenses, findings that were in agreement with his current state. There were also high scores on affiliation, pseudo-altruism and task-orientation defenses, and these mature attributes were expressed during gameplay: Paul played MMORPGs heavily, with an emphasis on the ‘healer’s’ role, willing to help others in the team and able to grind his way into game payoffs. Since Paul had little motivation to enter treatment or seek change, the treatment protocol was initiated very soon after a brief discussion of his history and his DSQ test results. Paul found his GAI sessions enjoyable and approached them with curiosity, being willing to complete them and showing a very rich imagination, being able to fill in his description with a considerable level of detail and lack of resistance. Paul produced striking images of desolation when cued to proceed with exploring semi-structured environments: an inner exploration led to a huge, abandoned library, set against a post-apocalyptic background, a metaphor for his high hopes of academic achievement in the past, now abandoned. Gradually, hopeful images begun to emerge, yet he refused to follow through with any conclusions that would entail him being more involved with this RL surroundings. In an open-ended scenario Paul found himself being chased by an aggressive gorilla away from a Mayan temple, depicting his aggressive tendencies that plagued his emotional relationship, although he managed to outrun it. He descended in a cave and faced an evil white dragon, a symbolism of his desire to escape from life’s pains and issues, a desire that was not beneficial for him. A traditional healer signified the emergence of hope and an arduous hike leading to a monastery on top of a tall hill signified his attempt to gain insight, regardless of effort. His ambivalence regarding his treatment gradually resolved and he became less guarded to outside events, eventually establishing a new romantic relationship after working through his complicated relationship with his mother, a person whom he had vowed a few years ago never to physically assault despite her misbehavior, a decision that in his eyes contrasted deeply with his physical assault against his earlier romantic partner. Paul gradually processed his loss of a perfect image that he had meticulously built the years prior to his episodes and started living for the moment, striving to constantly improve himself, as he always did, minus the goal of asserting dominance as he used to strive for. He put an end to smoking marijuana and greatly reduced his time spent online and especially gaming, rejoining his studies, and trying hard to make his new relationship work. He recognized that gaming and marijuana were escapist practices and set a new goal to respond to new challenges, especially in his relationship, by not succumbing to those practices. His success became a source of pride in himself versus his earlier feelings of inferiority vis-à-vis an idealized perfect Self.

What is especially noteworthy in the case of Paul is the comorbidity with bipolar disorder and marijuana use. Both definitions of IGD available by the American Psychiatric Association and the World Health Organization clearly stipulate that bipolar disorder should be ruled out before establishing the diagnosis, comparable to the notion that a manic bipolar patient should not be viewed as addicted to gambling or chemical substances, if they are abused exclusively during his manic episodes. However, in this case, Paul’s involvement with gaming predated his diagnosis with bipolar disorder, and more importantly, it was now a conscious choice of escaping a depressing reality where his dreams appeared lost and his future nullified as a result of him being diagnosed with a mental disorder. While there are a number of cases where disordered gaming is indeed linked to a manic episode, and the behavior would subside outside the episode, this was not the case here. For Paul, gaming was the sole remaining field where he could still chase perfection and it alleviated his depressive disposition. On the contrary, during his manic and his depressive episodes, his gaming subsided significantly since he could not concentrate. The same was true for his marijuana use, which mostly led to exacerbating his symptoms of depression rather than be exacerbated by manic elatedness. One should also stress that it is counter-indicated to journey with imagery in active psychotic disorders [35] and it is near impossible to do so during manic episodes. Paul’s journeys were very vivid, but he was able to frame them well and did not show mental fatigue or anguish; in any case, an experienced practitioner is needed to secure against any adverse outcomes.

## 4. Discussion

### 4.1. Application of the Treatment Plan

Although IGD has been viewed nearly unequivocally as a ‘behavioral’ disorder, cases of severe character pathology are frequent [26] and require alternate approaches, since the issues that patients face are deeply entrenched and present numerous obstacles, if approached with a typical brief treatment approach. Patients who have failed repeatedly in attempting to self-regulate and are stalled both in practical aspects of their lives but also developmentally, failing to sustain meaningful personal relationships and fulfilling occupations, will benefit from a psychoanalytically guided treatment approach.

Our conclusions from therapy sessions were that these patients have rich fantasy lives and are unlikely to simply give up on their megalomaniacal delusions of omnipotent control in order to return to their mundane ‘real’ lives and face everyday problems. These problems overwhelm their limited capacity for self-soothing and are avoided. However, the guided imagery approach meets them halfway and addresses their issues symbolically, which is less threatening than the prospect of direct self-exposure. The existence of archaic defenses seen in these two cases is typical of narcissistic disorders and the biggest obstacle to treatment. Placing the subject at the center of a guided imagery scene placates the narcissistic need for attention, since it amounts to the patient being the principal actor, around whom the whole play revolves. Although the patient enters the setting apprehensively, curiosity emerges on what comes next, which carries over to healthy curiosity on one’s own psychological makeup; the patients are themselves perplexed as to their root cause of their stalling and have tried starting out on the journey towards maturity only to be shunted due to their lack of inner structure able to withstand developmental stress. The patient will become more involved with therapy if he/she feels that it provides a meaningful explanation to his/her experience.

A separate issue that is not addressed during both case reports relates to the nature of the ersatz selfobject itself that makes it addictive; in all addictions the addict will shift from various chemical substances to more favorite ones, as noted by Taipale [23]. This discussion is sorely lacking in the IGD paradigm. There are a few reports pointing to MMORPG games being more addictive [37,38] and this is in agreement with our clinical experience. MMORPGs combine a chance to excel among peers, have simple and stable rules, never-ending play, lack of the sense of passage of time and a fantasy world far away from the trivialities of real life. These characteristics are inducive of the hypnogogical state of consciousness described by Ulman and Paul. These negative aspects are discussed with the patient who is then encouraged to reach his own conclusions as to the role that MMORPG play serves and find alternate pathways to fulfilling that role. Dropping out of this genre needs to become a conscious decision that the patients reach by their own accord. A relapse is seen as a helpful reminder of what lies within, and the steps still needed to reach a more mature Self.

### 4.2. Special Issues in Treatment

The two cases were selected as examples of issues which are not frequently met during IGD therapy, that of strong resistance during the imagery sessions and that of comorbidity with a psychotic disorder. Marijuana use was also present, yet this comorbidity is relevantly common with IGD. This is not to say that it does not affect its course. In the case of George, marijuana use numbed his anxiety and made it less likely to seek treatment. This was also true in the case of Paul, and in addition, marijuana exacerbated his symptoms of psychotic depression. Hence in any case, this comorbidity should also be addressed during treatment, as serving a similar role of an ersatz selfobject as gaming.

The types of games that the addicted players play typically are MMORPGs or other games with some teamplay, such as online shooters. The choice of game and character type within the game more often than not depends on what the player perceives as being easier to excel in. In the words of ‘Paul’, ‘I wanted to be a good player in one or two mainstream games, so that I would be able to meet people, ask them what they play, and demonstrate my talent.’ The development of a cohesive self takes place along three axes: the grandiosity axis, the idealization axis, and the alter ego–connectedness axis [27,32]. This sentence perfectly echoes the stunted development in the grandiosity axis, where the self-esteem depends on external sources of positive feedback, but also the stunted growth in the alter ego–connectedness axis, where team play serves to alleviate the sense of psychic isolation. Improvement during the game would be a motivating factor for playing, but eventually, if the player did not achieve the status of a better-than-average player, then he/she would move on to another game. This need to self-validate through gaming explains the choice of MMORPGs since they combine competitiveness, a team system, and a sense of constant improvement via leveling up. Self-validation is thus crucial in addressing gaming addiction and the patient should be redirected to a fulfilling activity, preferably in a group setting, while refraining from indulging in gaming,

The advent of micro-transactions and loot boxes, as described above, has blurred the boundaries between gaming and gambling. There was a small number of players who reported for treatment in the ‘student outpatient Service for the treatment of problematic use of personal computers and the internet’ after spending considerable amounts of money on in-game purchases. These purchases are of items that ensure their rapid progression through the ranks or diversify them from the other players (e.g., an ornate weapon or special ‘skin’ for the n-game character). The funds that they use may not be disposable, demonstrating that their purchases carry a deeper psychological meaning than simple recreation, as they would initially justify them. When probed deeper, a sense of resentment emerges, even rage, comparable to separation angst from a self-object. Although the actual in-game value of their purchases amount to some moderate gains at best, they cannot accept the notion of limiting themselves by not making these purchases. During their treatment it was revealed that at the core of this issue was the inability to accept one’s limitations, again a marking of stunted growth in the grandiosity axis. Loot boxes present a parallel to gambling and patients who spend time or money on attaining them usually also gamble to some degree, more often casually. Treatment sessions showed that ‘luck’ in general has a small place in IGD; players will recognize that the more sessions you complete, the less likely it is that you will be seriously affected by adverse coincidences. ‘Chasing one’s luck’ as such is seen as a rookie’s approach; one of the main draws to online gaming is its predictability versus the chaotic nature of the world outside. If you put in the hours and the effort, then you will be justified in expecting results and noticing the improvement. If the addicted player loses a session to an opponent who seems to be favored by chance, then a feeling of intense rage emerges, as if the very nature of the universe was threatened. Narcissistic rage in this case stems from a sense of the opponent essentially stealing the libidinal investment that the patient made onto his involvement with the selfobject. The patient cannot come to terms with this perceived injustice and treats the opponent as if he/she willingly inferred this damage onto him. However, since there is nothing that can be done about it, the player explodes in narcissistic rage, much like a child ‘veers back in forth between enraged protests at the imperfection of his grandiose self and angry reproaches against the omnipotent self-object for having permitted the insult’, as Kohut noted [33].

### 4.3. Limitations of the Treatment and Future Directions

Although the treatment plan does not assume that the patient is highly motivated for change, a moderate degree of cooperation and openness is required, since the patient may sabotage the course of treatment, as seen in the case of George being reluctant to fully disclose his experience during the sessions and his marijuana use. The most important point, however, is that the patient needs to have intact reality testing. This presumes that the patient will not be intoxicated with any substance during the session and will not be undergoing a psychotic episode. An extension of the former is that a patient who is deeply emersed in the online gaming experience will have trouble concentrating during the sessions and perhaps more importantly, processing the content of the session afterwards. In this instance, it is strongly advised that the patient makes an effort to curtail his gaming sessions to a moderate degree before he/she enters treatment.

Future research should focus on the long-term outcomes of this treatment modality. A point of interest would be the better integration with other treatment modalities; ideally the patient should be referred to the appropriate intervention following a standardized diagnostic assessment common for at least three potential interventions: a cognitive-behavioral approach for patients with limited character pathology and less contact with psychiatric services in the past, the psychoanalytically guided approach for patients with character pathology and other co-morbidities but with intact reality testing and a supportive approach for patients with other severe co-morbidities (e.g., chemical addictions or psychotic disorders) with limited reality testing.

## 5. Conclusions

Psychoanalytically guided treatment for IGD can be very effective in cases with character pathology who present unique challenges for the therapist. These cases are more often as the patient’s age increases, pointing to an early developmental arrest that manifests itself as the environmental demands increase. With IGD already identified as a public health risk, it is important to provide theory-driven alternatives to existing treatment approaches in order to provide the optimum standard of care, especially with regards to patients with long-standing character pathology and comorbid psychiatric disorders.

## Data Availability

Not applicable.

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
