# Peer review of "A Psychoanalytic Approach to Internet Gaming Disorder"

_ijerph, 2023, doi:10.3390/ijerph20156542_

Round 1
Reviewer 1 Report
This is a very interesting study that explores a treatment method often unheard of when seeking treatment for Internet Gaming Disorder (IGD). The author effectively provide evidence for the reader in regards to the value of a psychoanalytical approach to IGD treatment in the later half of the introduction section, and the 2 cases appropriately present the value of this treatment approach.
My concerns with this article will be as follows
1. the initial few lines of the article (30 to 41) include a lot of opinions but not a lot of facts about IGD. Authors can consider using prevalence data for IGD, the rapid increase in global internet users and the internet gaming community, the high number of individuals engaging in MMO RPGs etc. to emphasize the importance of further research in IGD rather than listing the opinion of a few individuals.
2. Authors can consider exploring whether this approach will be useful for other internet-based games like online shooters, or Minecraft. Playing as part of a team is essential in online games, and while some of these games do not have the creative freedom offered by RPG games to create an individuals character; many of these games have a character archetype (e.g. a medic in a combat shooter) that a specific individual may gravitate towards.
3. Many online games also have a component of "luck". some games randomly assign you to a team, so it is up to you to find a way to work with the new team. Other games offer "loot" on completing objectives that range from something worthless to something extremely rare. many use "real world" money to buy "loot boxes" that offer an opportunity. How do the authors think their treatment can be suited to address these issues?
I enjoyed reading this paper and hope that this approach is further utilized to help patients struggling with this terrible malady.
Author Response
Thank you for the opportunity to amend out manuscript following your very useful suggestions. A detailed response to all comments follows:
-------------------------------------------------------------------------------------------
This is a very interesting study that explores a treatment method often unheard of when seeking treatment for Internet Gaming Disorder (IGD). The author effectively provide evidence for the reader in regards to the value of a psychoanalytical approach to IGD treatment in the later half of the introduction section, and the 2 cases appropriately present the value of this treatment approach.
My concerns with this article will be as follows
1. the initial few lines of the article (30 to 41) include a lot of opinions but not a lot of facts about IGD. Authors can consider using prevalence data for IGD, the rapid increase in global internet users and the internet gaming community, the high number of individuals engaging in MMO RPGs etc. to emphasize the importance of further research in IGD rather than listing the opinion of a few individuals.
Response: Thank you for your comment, the Introduction section has been expanded to include definition, prevalence and key aspects of IGD (lines 25-60) while presenting the current state regarding online gaming, MMORPG gaming and aspects like microtransactions and loot boxes (lines 61-75)
2. Authors can consider exploring whether this approach will be useful for other internet-based games like online shooters, or Minecraft. Playing as part of a team is essential in online games, and while some of these games do not have the creative freedom offered by RPG games to create an individuals character; many of these games have a character archetype (e.g. a medic in a combat shooter) that a specific individual may gravitate towards.
Response: Thank you for your comment, this notion has been expanded in the Discussion section which was rewritten to include special issues in treatment (4.2), limitations and future directions (4.3).
3. Many online games also have a component of "luck". some games randomly assign you to a team, so it is up to you to find a way to work with the new team. Other games offer "loot" on completing objectives that range from something worthless to something extremely rare. many use "real world" money to buy "loot boxes" that offer an opportunity. How do the authors think their treatment can be suited to address these issues?
Response: Thank you for your comment, this is indeed a serious concern, it has been briefly presented in the Introduction section and expanded in the Discussion section (4.2)
I enjoyed reading this paper and hope that this approach is further utilized to help patients struggling with this terrible malady.
Response: Thank you for your kind words and your insightful comments.
Reviewer 2 Report
The paper provides some valuable insights into psychoanalytical approach to ICG, and the treatment plan is clearly described. However, there are some points that need to be addressed.
The introduction must be improved as it lacks information/citations at times to support the statements/arguments, e.g. in relation to the main age groups and motives of internet gaming, a concise but clear introduction about existing approaches and psychoanalytical approach along with its potential advantages. This applies to other sections as well, e.g. "gaming-addicted patients have a penchant for daydreaming and especially escapist fantasy life..." - but no citation provided for this.
Introduction/description about IAD is missing.
Where did the cases of patients come from/who provided these cases - it needs to be clarified including the ethics/confidentiality issues.
Discussions need to be further established in relation to in-depth reflections and recommendations for future research and practice.
Author Response
Thank you for the opportunity to amend out manuscript following your very useful suggestions. A detailed response to all comments follows:
The paper provides some valuable insights into psychoanalytical approach to ICG, and the treatment plan is clearly described. However, there are some points that need to be addressed.
- The introduction must be improved as it lacks information/citations at times to support the statements/arguments, e.g. in relation to the main age groups and motives of internet gaming, a concise but clear introduction about existing approaches and psychoanalytical approach along with its potential advantages. This applies to other sections as well, e.g. "gaming-addicted patients have a penchant for daydreaming and especially escapist fantasy life..." - but no citation provided for this.
Response: Thank you for your comment, the Introduction has been re-written following your suggestions to include the additional information with additional references where needed.
- Introduction/description about IAD is missing.
Response: Thank you for your comment, this information has been added lines 25-60 and the current state of online gaming along with some special issues as suggested by Reviewer #1 added in lines 61-75
- Where did the cases of patients come from/who provided these cases - it needs to be clarified including the ethics/confidentiality issues.
Response: Thank you for your comment, This information has been added in the beginning of section 2.3
- Discussions need to be further established in relation to in-depth reflections and recommendations for future research and practice.
Response: Thank you for your comment, the Discussion section has been expanded to include to include special issues in treatment (4.2), limitations and future directions (4.3).